# Electrospun Membrane Based on Poly(L-co-D,L lactic acid) and Natural Rubber Containing Copaiba Oil Designed as a Dressing with Antimicrobial Properties

**DOI:** 10.3390/antibiotics12050898

**Published:** 2023-05-12

**Authors:** Marcelo Formigoni Pinto, Bruna V. Quevedo, Jessica Asami, Daniel Komatsu, Moema de Alencar Hausen, Eliana Aparecida de Rezende Duek

**Affiliations:** 1Mechanical Engineering Faculty (FEM), State University of Campinas (UNICAMP), Campinas 13083-860, São Paulo, Brazil; 2Post-Graduation Program in Materials Sciences (PPGCM), Federal University of São Carlos (UFSCar), Sorocaba 18052-780, São Paulo, Brazil; 3Laboratory of Biomaterials, Faculty of Medical Sciences and Health (FCMS), Pontifical Catholic University of São Paulo (PUC-SP), Sorocaba 18030-070, São Paulo, Brazilmahausen@pucsp.br (M.d.A.H.); 4Post-Graduation Program of Biomaterials and Regenerative Medicine, Surgery Department, FCMS, PUC-São Paulo, Sorocaba 18030-070, São Paulo, Brazil

**Keywords:** natural extract, PLDLA, natural rubber, antibacterial, antibacteriostatic, electrospun membrane

## Abstract

Drug delivery systems of natural antimicrobial compounds, such as copaiba oil (CO), have become relevant in the scientific community due to the recent prevalence of the public health complications related to antibiotic resistance. Electrospun devices act as an efficient drug delivery system for these bioactive compounds, reducing systemic side effects and increasing the effectiveness of the treatment. In this way, the present study aimed to evaluate the synergistic and antimicrobial effect of the direct incorporation of different concentrations of CO in a poly(L-co-D,L lactic acid) and natural rubber (NR) electrospun membrane. It was observed that CO showed bacteriostatic and antibacterial effects against *S. aureus* in antibiogram assays. The prevention of biofilm formation was confirmed via scanning electron microscopy. The test with crystal violet demonstrated strong bacteria inhibition in membranes with 75% CO. A decrease in hydrophilicity, observed in the swelling test, presented that the addition of CO promotes a safe environment for the recovery of injured tissue while acting as an antimicrobial agent. In this way, the study showed strong bacteriostatic effects of the CO incorporation in combination with electrospun membranes, a suitable feature desired in wound dressings in order to promote a physical barrier with prophylactic antimicrobial properties to avoid infections during tissue healing.

## 1. Introduction

Large or chronic skin injuries can represent a severe health risk to the human body since skin functions may be impacted. Important issues, such as thermal insulation, body fluid loss, and bacterial infections, must be hindered. Due to epidermis necrosis, clinical treatment focuses on preventing infections to ensure full dermis recovery. Therefore, bacterial infections are a serious threat to the healing process and, consequently, to public health all over the world. The emergence of new super-bacteria, resistant to assorted drug classes, is challenging, not only in the infection treatment but creates efforts to develop an unfriendly environment for its growth [1]. However, *Staphylococcus aureus* is among the most prevalent biofilm-forming bacteria in skin infections [2]. The biofilm formation provides protection for bacteria through a physical barrier against drugs, i.e., creates a conducive environment to their survival and proliferation [1]. Thus, new approaches and devices must be studied and developed to act in the prevention of bacterial infections.

Current designs of curative devices are looking to not only act as a bare physical barrier membrane but mainly as an active dressing that could induce a tissue response and/or prevent infection. Additionally, a dressing that could act effectively as a pharmacological delivery directly to the site of action could minimize or avoid side effects and prevent microorganism growth [3]. To reach an effective release, the drug and the material used for delivery purposes must be processed to perform a sustained controlled release while maintaining the material property. The materials that are being used nowadays for this design include nanogels, microspheres, hydrogels, and electrospun structures [4]. 

Electrospun nanofiber-based wound dressings have arisen as one of the most promising therapeutic solutions for wound healing management, since they exhibit structural features that are quite similar to those presented by the native skin’s extracellular matrix (ECM). The electrospinning technique has been used in research for drug encapsulation since electrospun *scaffolds* present an extensive porous fibrillate surface, which supports drug penetration in the polymer structure [5], while the big surface area favors the high absorption of drugs due to the wide surface available [6]. 

In this context, several studies point to the versatility of electrospun membranes that act systematically in the targeted delivery of various substances, including natural compounds with pharmacological and antimicrobial properties. In this way, electrospun membranes composed of poly(vinyl)pyrrolidone were able to act effectively in the release of herbal extracts [7], while anti-inflammatory and natural antibacterial agents were incorporated into the nanofibrous membrane of poly(vinyl alcohol) (PVA) and lysine [8]. The electrospun asymmetrical membrane (EAM), composed of silk fibroin, poly(caprolactone), and hyaluronic acid (HA), was loaded with a herbal antimicrobial agent (Thymol) [9]. The chitosan/gelatin electrospun nanofiber was able to release cinnamon extract (CE) through sustained release [10]. Therefore, it is well described that there is consensus in the scientific community on studies involving the incorporation of natural therapeutic compounds into electrospun polymeric membranes, aiming at applications in skin dressing [11]. 

In addition, it is worth mentioning copaiba oil (CO). CO is a natural compound extracted from plants of the genus Copaifera, a native species from the Brazilian Amazon region. It is found in different species, such as *Copaifera officinalis* L., which was used in this study. CO has important biological properties for biomedical applications, such as analgesic, antiparasitic, anti-inflammatory, antioxidant, and antimicrobial activity [12]. Furthermore, CO shows antibacterial [13] and bacteriostatic [14] properties, important properties to prevent infections of high complexity. The bioactivity of CO is mainly due to the presence of flavonoids, diterpenes, and sesquiterpenes, such as β-bisabolene, β-caryophyllene, α-copaene, and α-humulene [2,15].

The solubilization of the polymeric matrix to be electrospun is the main technique used [16]. However, incorporating drugs in this process could be challenging when the bioactive compound is solubilized together with the polymeric matrix before the electrospinning process [17,18]. A disadvantage has been reported in this method due to a loss in drug activity, once the solvent used in the electrospinning process could denature some types of bioactive molecules [3]. In this work, the electrospinning procedure was used to create a membrane followed by the addition of CO after electrospinning with a two-step method. The first step was the simultaneous electrospinning of poly(L-co-D,L lactic acid) (PLDLA) and natural rubber (NR) of both polymers, generating a mixed-pattern membrane with these two materials. The choice of these polymers is due to their inherent properties for application in the development of new biomaterials but considering, at first, the purposes a low-cost device and a material with easy degradability properties due to environmental approaches. In this way, the PLDLA (70/30) consists of an amorphous aliphatic polyester with the following biological properties: compatible, absorbable, and degradable [19]. On the other hand, the NR biopolymers from *Hevea brasiliensis* rubber trees are composed of long chains of poly(*cis*-1,4-isoprene), and due to such properties, they provide increased malleability and flexibility to the PLDLA due to its elastomeric features. Moreover, it is biocompatible and stimulates angiogenesis, accelerating healing and tissue repair [20]. The second step was performed with the pure CO incorporated through full immersion followed by drying the surface of the electrospun membrane. This method guaranteed the bioactive oil properties and showed promising results to prevent biofilm formation.

## 2. Results

### 2.1. Fourier-Transform Infrared Spectroscopy

The FTIR spectra of CO, electrospun PLDLA/NR membrane and electrospun PLDLA/NR membrane incorporated with 25, 50, and 75% CO are shown in Figure 1. The CO spectrum showed two intense peaks, observed at 2924 and 2850 cm^−1^, attributed to the symmetric stretching of the –CH bonds in alkanes and aldehydes, respectively. In addition, the peak at 1697 and 1646 cm^−1^ suggests carbonyl stretching. Moreover, the peaks at 1447, 1380, and 887 cm^−1^ indicate the bending vibrations of aliphatic CH_2_, C–O bond vibrations, and C=C bending, respectively [12,21].

The PLDLA/NR infrared spectrum presented the main bands of both individual polymers (PLDLA and NR). In this case, a peak was observed at 2924 cm^−1^, related to absorption bands corresponding to the symmetric (CH_2_) axial deformation of C-H bonds. At 1748 cm^−1^, a characteristic peak of polyesters can be observed, which is associated with the axial deformation of the carbonyl group (C=O). At 1450 cm^−1^, there is a peak related to the angular deformation of C-H bonds in the CH_3_ group. At 1081 cm^−1^, the characteristic polylactide bands corresponding to the =C-O stretching can be observed. At 1181 cm^−1^, there is a characteristic peak of the C-C bond [19]. The NR infrared spectrum presented bands of symmetrical and asymmetrical stretching of CH_2_ and CH_3_ groups between 2980 and 2850 cm^−1^ (2962, 2915 and 2850 cm^−1^). The peak at 1375 cm^−1^ is also related to -CH_3_ deformation [22]. The PLDLA/NR spectrum, incorporated with different CO concentrations, showed a difference when compared to the PLDLA/NR spectrum due the presence of weak-intensity peaks at 1646 and 887 cm^−1^. These peaks are related to CO. Moreover, no difference was observed between the PLDLA/NR with CO spectrum and the PLDLA/NR spectrum, indicating that polymers (PLDLA and NR) do not present structural changes or interaction between them. Furthermore, the non-peak change in the FTIR spectra (PLDLA/NR with CO samples and PLDLA/NR sample) indicates that there was no interaction between both polymers and CO molecules.

### 2.2. In Vitro Antimicrobial Activity

The antimicrobial activity of the electrospun PLDLA/NR membrane containing CO against *S. aureus* at 8 and 24 h of culture is shown in Figure 2. The control samples did not exhibit an inhibitory effect against *S. aureus*, except the commercial control of doxycycline. On the other hand, electrospun PLDLA/NR membranes incorporated with 25 and 50% of CO exhibited bacteriostatic and inhibitory effects at 8 and 24 h of culture, respectively. The inhibition zone exhibits a striking decrease from 8 to 24 h. In this case, electrospun PLDLA/NR membranes with 25% and 50% of CO showed inhibition zones that decreased from 15 to 6 mm and from 18 to 8 mm, respectively. The diffuse pattern of the inhibition zone observed after 24 h indicated that the CO bacteriological properties were limited, and the *S. aureus* started to present a late growth in the area that was previously related to the inhibition zone. As pointed out by Norcino et al. 2020 [21], the bacteriostatic behavior of CO can contribute to the maintenance of bacterial growth in a stationary phase, even in low concentrations. Similar results identified for 25 and 50% were not followed when 75% of CO was added to the PLDLA/NR. After 8 h, the inhibition zone depleted the bacterial growth in almost all plate areas. However, after 24 h, the inhibition zone was equivalent to the concentrations of 25 and 50%. The differences in halo formation from 8 to 24 h reveal a loss in the inhibition zone of an average of 60% that led to a inhibition zone from 38 mm to 13 mm. On the other hand, CO incorporation to the PLDLA/NR membrane was not able to inhibit *E. coli* in all concentrations tested [23].

### 2.3. Zeta Potential Measurement

The influence of CO presence on the surface charge properties of the electrospun PLDLA/NR membrane was analyzed through surface zeta potential (ζ) as a function of pH changing, as shown in Figure 3. 

All analyzed membranes showed negative potentials ζ in all pH ranges. Thus, the electrospun PLDLA/NR membrane had a surface charge close to ₋15 mV in all pH ranges analyzed. Therefore, this is due to the chemical group’s presence of both polymers that compound the electrospun membrane, such as the carboxylic acid present in the polylactide (PLDLA) and in the remaining proteins and phospholipids that involve the colloidal system of NR [24,25]. 

On the other hand, the PLDLA/NR membrane with different concentrations of CO showed a pH-dependent behavior. Thus, after adding CO on the PLDLA/NR membrane (PLDLA/NR membranes +CO), there was a slight increase in the surface zeta potential value to approximately −10 mV at pH values below 5.5 when compared to the PLDLA/NR membrane. After the pH increased (pH > 5.5), there was a gradual decrease in the surface zeta potential value for all membranes with CO analyzed. In this case, PLDLA/NR membranes with 25, 50, and 75% CO presented (pH = 9) surface zeta potential values ζ −22, −27, and −29 mV at the end of the experiment, respectively. Therefore, the increase in negative charge on the membrane surface after CO was added could be related to the presence of ionizable groups on the polymer surface, such as resin acid that is present in CO composition [21].

### 2.4. Biofilm Formation through SEM

SEM images of the PLDLA/NR electrospun membrane without CO and with 25, 50, and 75% of CO, after 24 h of incubation with *S. aureus*, are shown in Figure 4. The bacterial biofilm formation was observed on the entire surface of the electrospun PLDLA/NR membrane without CO (Figure 4A,B). The bacteria presence either dispersed or agglomerated, indicating the colony formations along the fiber. *S. aureus* showed typical spherical morphology of coccus bacillus, with a diameter ranging from 0.37 to 0.46 µm (±0.072), corroborating the results described by Medveďová and Valík (2012) [26]. On the other hand, the minimal CO incorporation in the electrospun membrane entailed a drastic decrease in the bacteria number that could adhere to the membrane (Figure 4C,D). The incorporation of 25% CO was able to cause damage to the *S. aureus* cellular wall, evidenced by amorphous irregular bacterial fragments of their typical morphology. However, the incorporation of CO higher concentrations (50% and 75%) exhibited an antibiofilm effect, as the presence of bacterial cells along the fibers was not observed (Figure 4E,H). The aspect of electrospinning observed through the fiber morphology showed a pattern rich in recesses and pores, generating a favorable factor of a privileged microenvironment in the formation of microorganisms. Thus, the presence of CO was fundamental in not favoring this characteristic.

### 2.5. Crystal Violet Assay

The Crystal Violet technique is described in the literature to quantify bacterial biofilm formation, as already observed by Shukla and Rao (2017) [27]. After 24 h of *S. aureus* cultivation in an electrospun PLDLA/NR membrane, turbidity is observed in the control group (electrospun PLDLA/NR membrane without oil) and 25% CO group. The 50% and 75% CO group showed a more translucent culture (Figure 5), i.e., a smaller bacterial presence due to the release of oil concentrations in the bacterial culture medium.

The results observed via Crystal Violet staining (Figure 5) indicated that concentrations of 50% and 75% of CO inhibited bacterial growth with a remaining alive rate of 17% and 10%, respectively. The electrospun PLDLA/NR membrane with 25% of CO showed a high biofilm bacterial growth rate of 82% as compared to the electrospun CO-free PLDLA/NR membranes that were considered to be at a growth rate of 100%, as quantified through Crystal Violet assay. Despite the striking differences between 0%, 50%, and 75% of CO in this assay, the remaining bacteria were not able to colonize the surface of the electrospun PLDLA/NR membranes as observed via SEM. The electrospun PLDLA/NR membrane with 25% of CO also showed this difference when compared to the higher CO percentages. This shows that the presence of CO in higher concentrations, especially with 75% of CO, resulted in less biofilm formation on the electrospun PLDLA/NR membrane, as well as the results obtained using the antibiogram and the SEM images.

### 2.6. Oil Copaiba Release Test

The release profile of the average of three samples presented standard deviation lower than 5% of CO incorporated at different concentrations into the electrospun PLDLA/NR membrane shown in Figure 6. A similar release behavior was observed for all CO concentrations tested. Initially, all membranes showed a burst release (inserted graph) region followed by a sustained release up to 180 h. Thus, electrospun membranes with 25, 50, and 75% of CO were able to release up to 1 h of testing approximately 4.0, 5.0, and 4.5 mg.mL^−1^ of CO, respectively. Such release behavior is associated with the CO molecules adsorbed on the material surface, which are easily released into the medium in which the analysis is performed [28]. However, even increasing the concentration of CO added to the electrospun PLDLA/NR membrane, there was no increase in the amount of CO released in the initial test time.

It was noted that after 180 h, there was a gradual increase in the amount of CO released. In this case, the electrospun membrane with 75% of CO showed a significant increase in the amount of CO released compared to other electrospun membranes with the lowest concentration of CO. The difference in release kinetics may be associated with the inherent morphological structure of the tangled and porous fibers present in electrospun PLDLA/NR membranes. Therefore, the morphological structure is responsible for maintaining the release profile for a longer time, regardless of the concentration.

In order to understand the mechanism associated with the release of CO from the electrospun PLDLA/NR membrane, the experimental data were fitted to Korsmeyer–Peppas and Higuchi kinetic models, as shown in Figure 7. Although models are typically adjusted for all interval time, the obtained correlation coefficient (R^2^) was not satisfactory. In their studies, Scaffaro et al. (2017) [29] pointed out that mathematical models used to describe drug release have limitations in predicting the “burst release” effect. In this sense, and corroborating the results of Scaffaro et al. (2017) [29] and López-Muñoz et al. (2022) [30], the release data were fitted to mathematical models and divided into two-time intervals, corresponding to the two release phases. Thus, it was observed that the separate modeling of the two-time intervals fitted effectively to the power law model. Furthermore, the power law provides us with the values of (n), which is the exponent that describes the release mechanism. When the value of n ≤ 0.5, drug release occurs through Fickian diffusion, but when 0.5 < n < 1, an anomalous transport occurs (non-Fickian), while the value of k represents the kinetic constant related to the properties of the release system.

Figure 8 presents the variation in weight by the means of three samples that presented a standard deviation lower than 2% in all samples, as related to the immersion time of the samples in PBS at 37 °C.

It was observed that the percentage of swelling decreased with the increase in CO in the electrospun PLDLA/NR membrane. The electrospun PLDLA/NR membrane showed greater swelling capacity than the electrospun PLDLA/NR membrane with 25, 50, and 75% of CO. Therefore, the CO-free electrospun PLDLA/NR membrane exhibited a maximum swelling around 158% after 48 h testing, while the electrospun PLDLA/NR membranes incorporated with 25, 50, and 75% of CO exhibited 70, 30, and 17% of maximum swelling before 1 h testing, respectively. Therefore, the decreasing swelling value when CO concentration increases on the electrospun PLDLA/NR membrane can be attributed to the hydrophobic properties of the CO [26]. After reaching the maximum swelling, the membranes incorporated with CO showed a small reduction in the percentage of swelling. It occurred due to the release of CO into the medium, with swelling remaining constant until the end of the test (168 h).

## 3. Discussion

The urge in the scientific field due to the concern of microbial resistance is determinant in public health. The development of materials that possess active properties to trigger an action are the main demand. Electrospinning associated with bioactive molecules has gained significant attention recently [31,32]. Previous studies with CO properties also indicate this natural oil as a potential antimicrobiological compound for infection prophylaxis [23]. Thus, in this work, the incorporation of three different concentrations of CO in electrospun PLDLA/NR membranes for biomedical purposes was investigated.

FTIR spectra showed the presence of CO in the electrospun PLDLA/NR membrane (Figure 1). This result indicates the presence of sesquiterpenes, such as β-bisabolene and β-caryophyllene, that are responsible for engendering antimicrobial activity to CO [21].

In the comparative analysis of microbiology (Figure 2), it was observed that all electrospun PLDLA/NR membranes incorporated with CO, within a period of 8 and 24 h, showed a relative decrease in antimicrobial activity. According to Tobouti et al. (2017) [23], the different species of Copaifera may differ both in chemical composition and in activity against microorganisms. This decrease may also be associated with the rapid antibacterial action present in plant extracts. Santos et al. (2008) [13] showed, in their studies, that CO, under 3 h of culture, obtained an effective and fast bactericidal activity against Gram-positive bacteria, followed by a bacteriostatic effect up to 9 h of culture. Our results are also corroborative with Santos et al. (2008) [13], since the electrospun membranes also showed a bacteriostatic activity against *S. aureus*. According to Pieri et al. (2012) [14], the bacteriostatic effect is due to the copalic acid present in CO. Furthermore, the highest concentration (75%) of CO in the electrospun PLDLA/NR membrane exhibited the formation of an inhibition zone after 24 h of culture. Rodrigues et al. (2020) [33] evaluated the antimicrobial activity of CO incorporated directly into mangarite starch films and obtained an inhibition zone of 9.75 mm against *S. aureus*, which was considered effective. Other studies have shown that there is no clear consensus related to the acceptable levels of plant substances in bacterial growth inhibition compared to conventional antibiotics [34].

According to Rodrigues et al. (2020) [33], the antimicrobial activity against *S. aureus* observed in Figure 2 for an electrospun PLDLA/NR membrane with 75% of CO is also associated with the teichoic and lipoteichoic acid present in the cell walls of Gram-positive bacteria. Such substances are responsible for assisting the diffusion of hydrophobic compounds, such as CO in bacterial cells, contributing to the antimicrobial effect and formation of inhibition zones. On the other hand, the resistance against *E. coli* can be attributed to the external lipopolysaccharide membrane present in Gram-negative bacteria, which hinders the permeability of CO in the polysaccharide layer, generating a resistant protective barrier to the herbal oil. 

The surface zeta potential ζ as a function of changing pH (Figure 3) also contributes to the results regarding bacteriostatic and inhibitory activity, as shown in Figure 2. The negative charges present on the surface of electrospun PLDLA/NR membranes with different CO contents can easily interact with the peptidoglycans (positively charged) that interfere in the outer membrane of Gram-positive bacteria. Therefore, the interaction between the charges present in the electrospun membrane (negatively charged) and in the bacteria (positively charged) causes the destruction of the cell membrane, leading to cell death [35]. 

The bacteriostatic effects of CO in the samples were clearly identified via SEM (Figure 4), where no bacterial colonies were identified. It is important to highlight that the electrospun material presents a randomly oriented morphology with an interconnected fibrillar network that allows for the exchange of molecules and ions through its pores, but this enriched-pore topographic network needs attention as a potential advantageous microenvironment that could favor microbiological colonization [36]. Thus, the CO incorporation in the electrospun material, presented here, created an important feature that can prevent the formation of microbiological niches during the period tested. 

Comparing all microbiological results, a dose-dependent effect was observed on the inhibition of the formation of a bacterial biofilm, evident in the Crystal Violet (Figure 5) results when compared to SEM (Figure 4) and the inhibition plates (Figure 2). On the other hand, the direct correlation of the release profile (Figure 6) with higher or lower bacterial inhibition (microbiological results) could not be performed with a confident comparison due to the relative equal release distribution over time. Therefore, the results prove that the electrospun membrane allows for the diffusion of CO, favoring the inhibition of bacterial activity. Studies by Guimaraes and collaborators (2016) [37] demonstrated that CO has bactericidal capacity for the *S. aureus* species, at a concentration of approximately 0.3125 mg/mL, preventing growth activity. Bacterial cultures in BHI broth after 24 h demonstrated that the presence and release of CO in the electrospun membrane were efficient for inhibiting bacteria, according to McFarland’s Nephelometric Scale. Studies developed by Zapata and Arcos (2015) [38] focused on the turbidity of bacterial culture, based on the McFarland scale, showing the concentration of bacteria through the presence of medium turbidity. This fact expresses the effectiveness both in the release and in the bactericidal capacity of CO in the culture medium. According to the mathematical fit applied to the release results (Figure 6), the model that was better depicted was Korsmeyer–Peppas (Figure 7A). In this case, all samples showed the release exponent (n) with a value less than 0.5 (n < 0.5), indicating that the release of CO exhibits a typical mechanism of Fickian diffusion, with the drug release being influenced by other effects than just simple diffusion. According to Wang et al. (2017) [39], the Fickian diffusion was predominantly caused by CO permeation through the fibers of the electrospun material and can also occur, to a lesser extent, through pore diffusion when in contact with the release medium

The set of results indicated via microbiological assay (Figure 2), SEM (Figure 4), and Crystal Violet (Figure 5) indicated a low potential of this material containing CO for the treatment of ongoing infections. However, it showed an excellent potential for prophylactic purposes to prevent the adhesion and proliferation of microorganisms such as *S. aureus*. Thus, the assays (SEM), microbiology in plates/tubes, and colorimetric entangled results that lead to the conclusion that the CO release presented a low bactericidal effect, while it showed a strong bacteriostatic action. Attempting to avoid potential pathogens is a matter of paramount importance in clinical approaches. The use of bacteriostatic dressings that could be applied to injured areas that are usually submitted to exposure to contaminated surfaces, such as hands and feet, is a desired feature to prevent the growth and colonization of Gram-positive bacteria that could lead to a late infection of a tissue under a healing process. 

Comparing the swelling results (Figure 8), with the given context, it is observed that the swelling decreases as the concentration of CO in the electrospun PLDLA/NR membrane increases. This is due to the inherent hydrophobic characteristics of CO that prevent the hydrophilic molecules of the PBS solution from penetrating the membrane and allow for its swelling. Such a condition is desirable when directed to the treatment of wounds because dressing with more hydrophilic properties could cause a tissue collapse with the material, leading, consequently, to a strong adhesion to the exudate in the injured region. In this way, the characteristics assigned to CO can strategically act as a physical barrier, while preventing bacterial growth in the dressing, promoting a restriction in the interaction with the injured tissue, so that it could undergo other interventions later along the recovery process. Future works related to different concentrations of CO in assorted modified presentations of the material could incorporate a complete novel finding related to the bacterial behavior in different forms of dressings. Taking into account that the primary goal is related to the prevention of biofilm formation, we emphasize that other bacterial types should be further addressed to be tested in future analysis. 

## 4. Materials and Methods

### 4.1. Membrane Preparation

Solutions containing 10% of PLDLA (70:30) (w/vol.) and 1.5% NR (wPLDLA/wNR) were prepared separately in chloroform (Synth). The solutions were electrospun simultaneously through a DBM Eletrotech equipment system, model EF 2B RT 0219. CO (*Copaifera officinalis* L.) 100% pure and density of 0.920 g/mL, obtained from DV-Manipulation Pharmacy and Natural Products, and were then dispersed in the electrospun membrane at concentrations of 25, 50, and 75% of CO through of dilution in ethyl alcohol solution (70%) and Tween 20, as adapted from Rodrigues et al. 2020 [33]. The membranes with CO were dried at room temperature for about 3 days and stored in a vacuum desiccator until further analysis.

### 4.2. Characterization

#### 4.2.1. Fourier-Transform Infrared Spectroscopy

The FTIR-ATR (Spectrum 65 spectrophotometer (Perkin Elmer, Waltham, MA, USA) was performed at room temperature in a range between 4000 and 500 cm^−1^, with a resolution of 4 cm^−1^ and 32 scans.

#### 4.2.2. In Vitro Antimicrobial Activity

##### Bacteria Preparation 

The standard strains of *Staphylococcus aureus* ATCC 6538, *Escherichia coli* ATCC 1122 were used as reference for the antibiogram assay and also for biofilm formation. The strains were maintained and cultured as described by Malhotra et al. (2019) [40]. The biofilm assay (for SEM and Crystal Violet analysis) was adapted from Lencova and collaborators (2021) [41].

##### Antibiogram Assay

For the antibiogram assay, membranes with 25%, 50%, and 75% CO were cut into a size of 5 mm in diameter, and 15 µL of CO was added to each membrane slice. They were dried at room temperature for 24 h and sterilized in germicidal UV light. Three different membrane controls were designed as follows: negative control (PLDLA/NR), vehicle control (PLDLA/NR with (70%) ethyl alcohol + Tween 20), and antibiotic control with doxycycline (PLDLA/NR with doxycycline). All samples were placed on a plate with BHI medium and *S. aureus* or *E. coli* bacteria were seeded, each experiment was reproduced four times, each one in duplicate for each group tested. Samples remained in culture for 24 h in a bacteriological incubator at 37 °C. After the cultivation time, macroscopic images were obtained of the inhibition halo.

#### 4.2.3. Zeta Potential Measurement 

Zeta potential measurements were performed on an electrokinetic analyzer for solid surfaces (SurPASS, Anton Paar, Graz, Austria) with an adjustable gap cell. The surface zeta potential was determined as a function of pH. The 0.001 M KCl electrolytic solution had its pH changed (3 to 9) by adding 0.05 M HCl or 0.05 M NaOH through the automatic titration unit of the instrument, which had a pH variation of 0.5 per cycle. The pH and conductivity of the electrolyte were monitored using pH and conductivity probes, and all experiments were performed at room temperature.

#### 4.2.4. Biofilm Formation via SEM

Samples were cut into squares of 1 cm^2^ and processed as described for 4.2.2.2 topic. For negative control, the CO-free electrospun PLDLA/NR membrane was used. *S. aureus* and *E. coli* bacteria were seeded in all samples containing 3 mL of Brain Heart Infusion (BHI) medium (Sigma©, São Paulo, Brazil) at a concentration of 1.5x10³ CFU/mL and cultured for 24 h in a bacterial incubator at 37 °C. After 24 h, under culture, samples were washed with sterile phosphate buffer solution (PBS) (Sigma©, São Paulo, Brazil) and fixed in paraformaldehyde solution (PFA) (Sigma©, São Paulo, Brazil) 4% (w/v) for 2 h, followed by three baths in PBS. Samples underwent ethanol dehydration at growing concentrations up to two baths of 100% (Labsynth©, São Paulo, Brazil). For each percentage, samples remained in the bath for 30 min. Chemically critical point dried with hexamethyldisilazane (HMSD) (Sigma©, São Paulo, Brazil) and alcohol (100%) in a ratio of 1:1 for 10 min. After a final bath in pure HMSD, the samples were dried at room temperature. Prior to the observations, samples were sputtered with a gold/palladium coating (30mA, 60s) and analyzed via scanning electron microscope (JEOL, JSM-6010LA, Tokyo, Japan). 

#### 4.2.5. Biofilm Formation via Crystal Violet Assay

In order to carry out the Crystal Violet assay, the samples were submitted to the formation of biofilm of *S. aureus* bacteria. Samples were prepared as indicated in 4.2.4 up to 24 h of cultivation. After this period, samples were transferred to a new plate and washed five times with sterile distilled water, dried at room temperature for 45 min, and followed by Crystal Violet staining at 0.1% (Exodus Scientific, São Paulo, Brazil) for the same period. Samples were washed and incubated with 200 µL of 96% (v/v) ethanol for 15 min at room temperature. The solution was resuspended and transferred to a 96-well plate for absorbance read at 595 nm in a colorimetric microplate reader (ELX800UV Biotek©, Winooski, USA). Bacterial culture was performed directly in the surface of the electrospun CO-free PLDLA/NR membranes (control sample, 0% CO). The absorbance values were transformed to percentage values through an equation (Equation (1)):(1)$$Biofilm\;growth\;(\%)=\left(\frac{Biofilm\;with\;CO}{Biofilm\;without\;CO}\right)\times 100$$

The samples (quintuplicate) were compared between groups using one-way ANOVA with post hoc Tukey multiple comparison test.

#### 4.2.6. Oil Copaiba Release Test

To evaluate the release profile, 1 cm^2^ samples of electrospun membranes incorporated with different concentrations of CO were immersed in 6 mL of ethyl alcohol solution (70%) and Tween 20, followed by incubation at 37 °C. Aliquots (3 mL) were collected and replaced with the same volume in a time interval from 0 to 376 h. The release of CO was performed in a UV spectrophotometer (Femto Cirrus 80, São Paulo, Brazil) at 264 nm. The quantification of CO released was calculated using the mean obtained via a triplicate, as compared to a standard curve containing known concentrations. The CO release profile was described by the power law of Korsmeyer–Peppas (Equation (2)) [29]:*M_t_*/*M*_∞_ = *kt^n^*(2)
where *M_t_* = amount of released CO at an arbitrary time *t*; *M_∞_* = initial amount of CO added to the electrospun sample; *k* = release rate constant; and *n* suggests the nature of the release mechanism.

Furthermore, the Higuchi model was also applied (Equation (3)) [30]:*M_t_*/*M*_∞_ = *kt*^0.5^(3)
where *M_t_* = amount of CO released at an arbitrary time *t*; *M_∞_* = amount of CO initially added to the electrospun sample; and *k* = release rate constant.

#### 4.2.7. PBS Swelling Test

The swelling test was performed in triplicate. Squared samples of 1 cm^2^ were immersed in 10 mL of 0.01 M phosphate saline buffer solution (PBS) and kept in a thermostatic bath at a temperature of 37 °C for 168 h. Samples were removed from PBS at pre-established times, gently dried, and weighed on a 10^−4^ g electronic scale. The swelling percentage (%) of each sample was calculated through its initial weight (W0) and its weight after absorption (Wt) using (Equation (4)):(4)$$Swelling\;(\%)=\left(\frac{Wt-W0}{W0}\right)\times 100$$

## 5. Conclusions

FTIR-ATR analysis confirmed the presence of varying concentrations of CO in the electrospun PLDLA/NR membranes. The incorporation of CO provided bacteriostatic and antibacterial properties to the membranes. Notably, the membrane with the highest concentration (75%) of CO exhibited the greatest antimicrobial effect against *S. aureus* among the electrospun membranes. Zeta potential analysis indicated that the addition of CO to the electrospun PLDLA/NR membrane generated a surface charge difference that resulted in bacterial inhibition against Gram-positive bacteria. Additionally, the SEM results revealed that all CO concentrations effectively prevented biofilm formation on the surface of the electrospun membranes. However, even though the inhibition zone assays on plates suggested a possible dose-dependent inhibition in growth, it was not possible to establish a correlation between the CO release kinetics and the degree of antimicrobial effect, since the Crystal Violet assay showed strong inhibition of bacterial growth in the electrospun PLDLA/NR membrane incorporated with 75% of CO but only a fair inhibition in bacterial growth in membranes incorporated with 25% and 50% CO. The incorporation of CO also promoted a gradual decrease in the swelling capacity of the electrospun PLDLA/NR membrane. This condition was attributed to an increase in CO content that made the membranes more hydrophobic. Therefore, the inherent antibacterial and bacteriostatic properties of CO make the electrospun PLDLA/NR membrane a promising dressing to enhance therapeutic prophylaxis and prevent infections caused by Gram-positive bacteria, since *S. aureus* is one of the most common bacteria associated with skin infections.

## Figures and Tables

**Figure 1 antibiotics-12-00898-f001:**
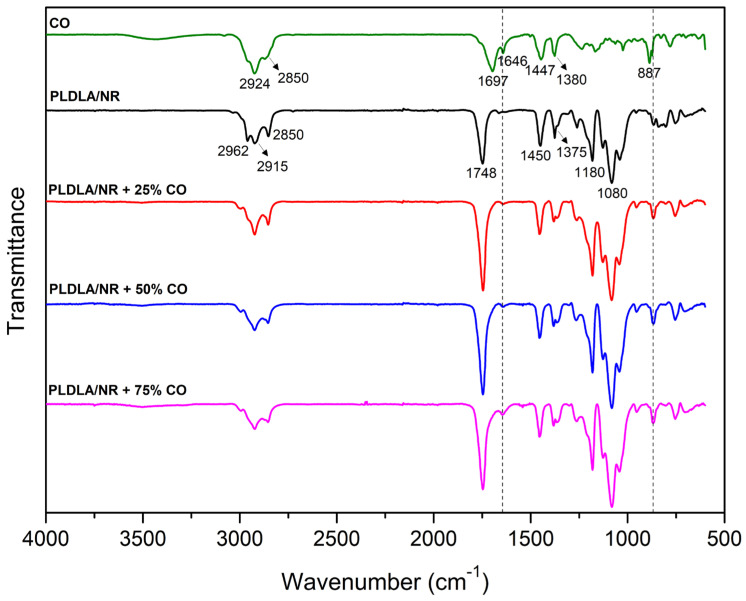
FTIR spectra of CO, electrospun PLDLA/NR membrane and electrospun PLDLA/NR membranes with 25% CO (PLDLA/NR+25%CO), 50% CO (PLDLA/NR+50%CO), and 75% CO (PLDLA/NR+75%CO). The peaks 1646 and 887 (highlighted by dotted lines) shows the stretching of CO.

**Figure 2 antibiotics-12-00898-f002:**
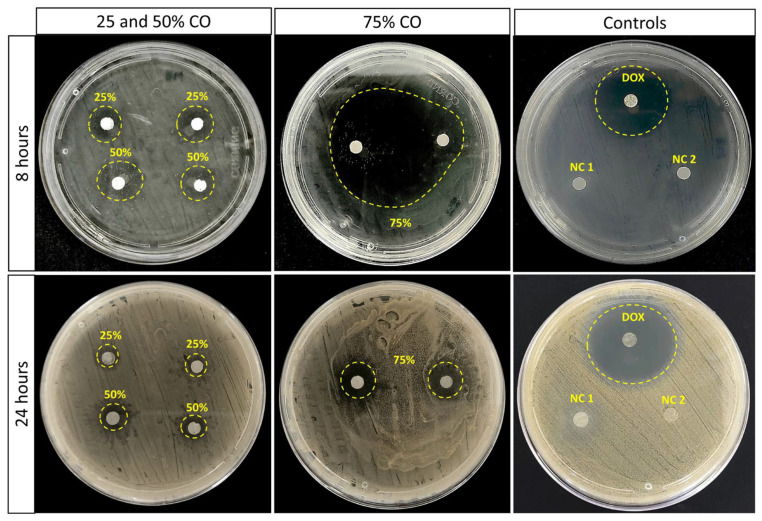
Antimicrobial activity of CO against *S. aureus* obtained via the agar diffusion test after 8 and 24 h of culture. Doxycycline (DOX) was used as a commercial antibiotics control, solution of ethanol with Tween 20 was used as a negative control (NC 1), as well as the disc of PLDLA/NR membrane CO free (NC 2).

**Figure 3 antibiotics-12-00898-f003:**
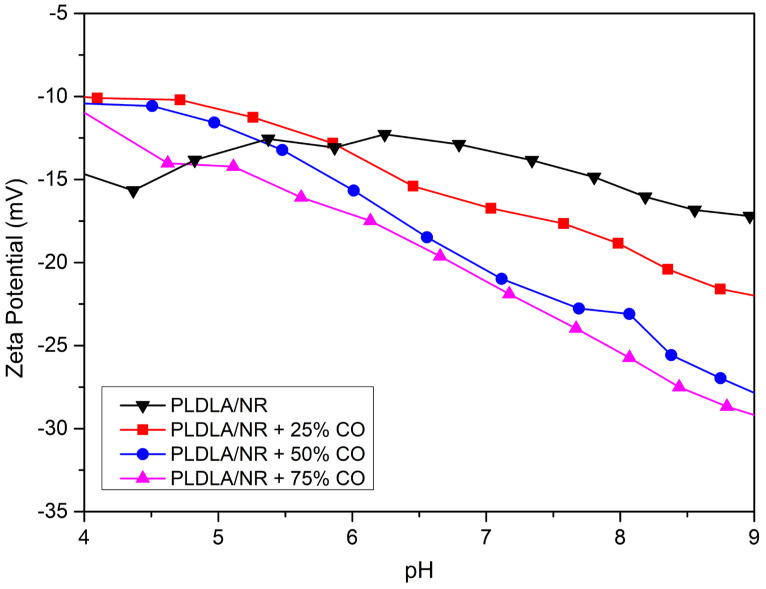
Surface zeta potential (ζ) in function of pH changing for electrospun PLDLA/NR membrane and electrospun PLDLA/NR membranes with 25% CO (PLDLA/NR + 25% CO), 50% CO (PLDLA/NR + 50% CO), and 75% CO (PLDLA/NR + 75% CO).

**Figure 4 antibiotics-12-00898-f004:**
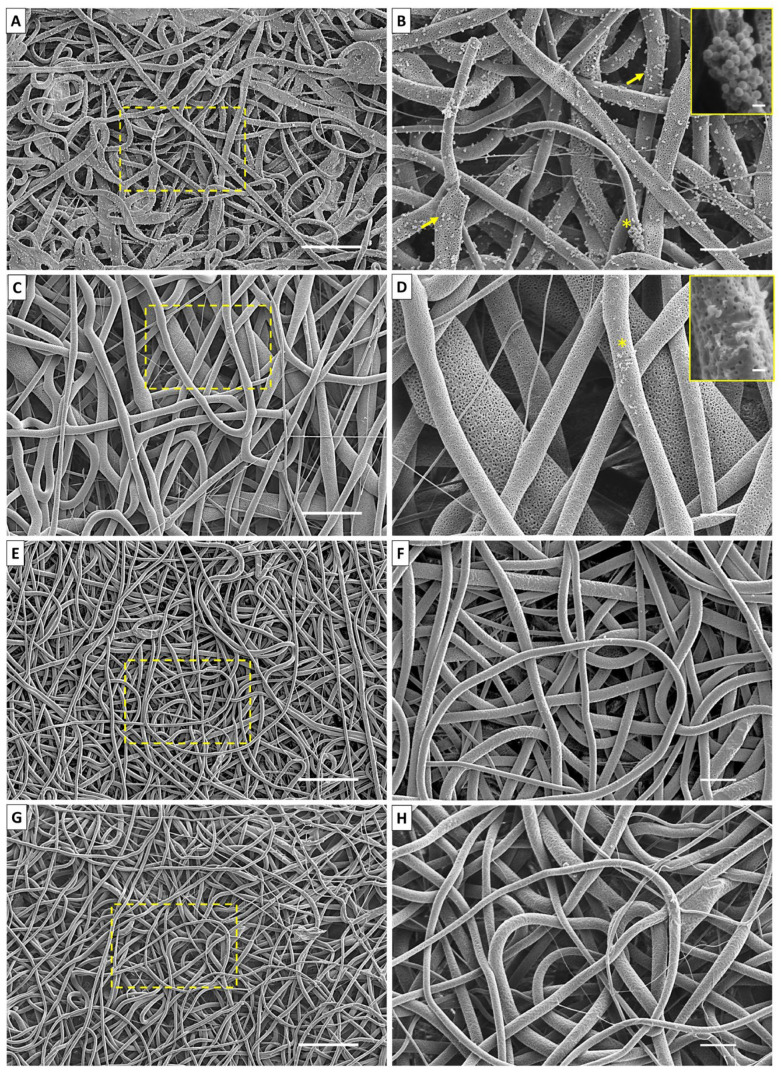
SEM images of PLDLA/NR membranes exposed to *S. aureus* during 24 h of incubation. (**A**) Electrospun PLDLA/NR membrane CO free (400x); (**B**) electrospun PLDLA/NR membrane without CO (1200x) (with bacteria (indicated by arrows) and colony of bacteria (*) indicated in the enlarged region (10,000x)); (**C**) electrospun PLDLA/NR membranes with 25% CO (400x); (**D**) electrospun PLDLA/NR membranes with 25% CO (with traces of bacteria (*) indicated in the enlarged region (10,000x)); (**E**) e (**F**) electrospun PLDLA/NR membranes with 50% CO (400 and 1200x); (G) e (**H**) electrospun PLDLA/NR membranes with 75% CO (400 and 1200x). Bars 50, 10, and 1 µm.

**Figure 5 antibiotics-12-00898-f005:**
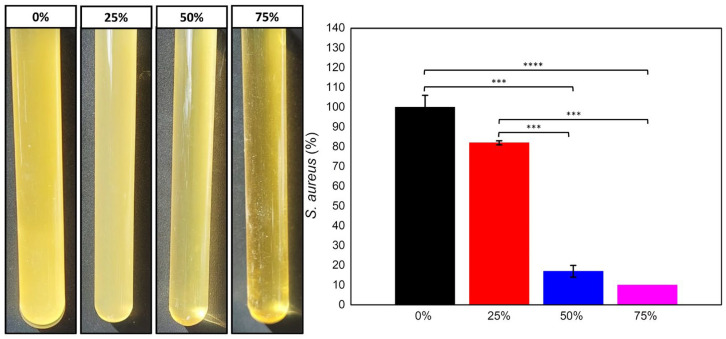
Illustrative image of the *S. aureus* culture turbidity in electrospun membrane oil-free (0%), 25%, 50%, and 75% of CO. Graph of Crystal Violet staining analysis of the percentage of bacteria present in the formation of biofilms on the electrospun membranes and the respective oil concentrations, where *** is *p* > 0.0001 and **** is *p* < 0.0001.

**Figure 6 antibiotics-12-00898-f006:**
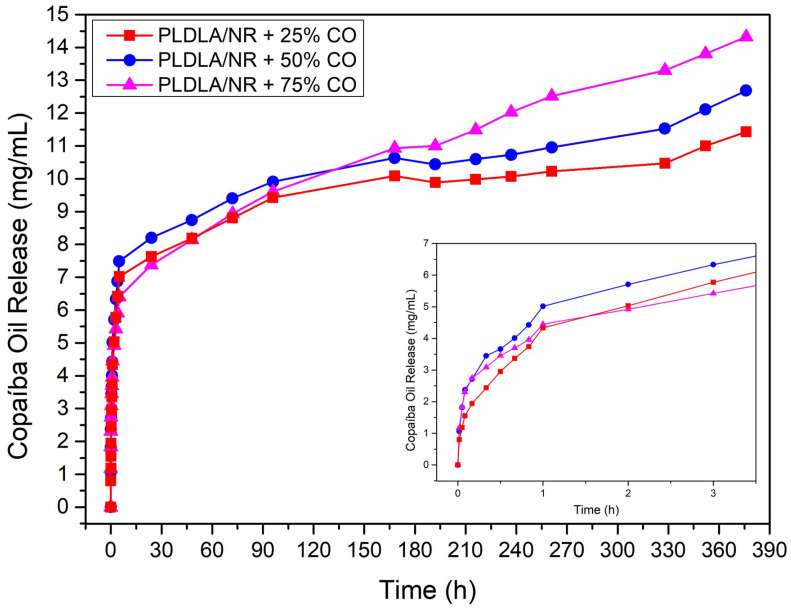
CO release profile of electrospun PLDLA/NR membranes with 25% CO (PLDLA/NR + 25% CO), 50% CO (PLDLA/NR + 50% CO), and 75% CO (PLDLA/NR + 75% CO).

**Figure 7 antibiotics-12-00898-f007:**
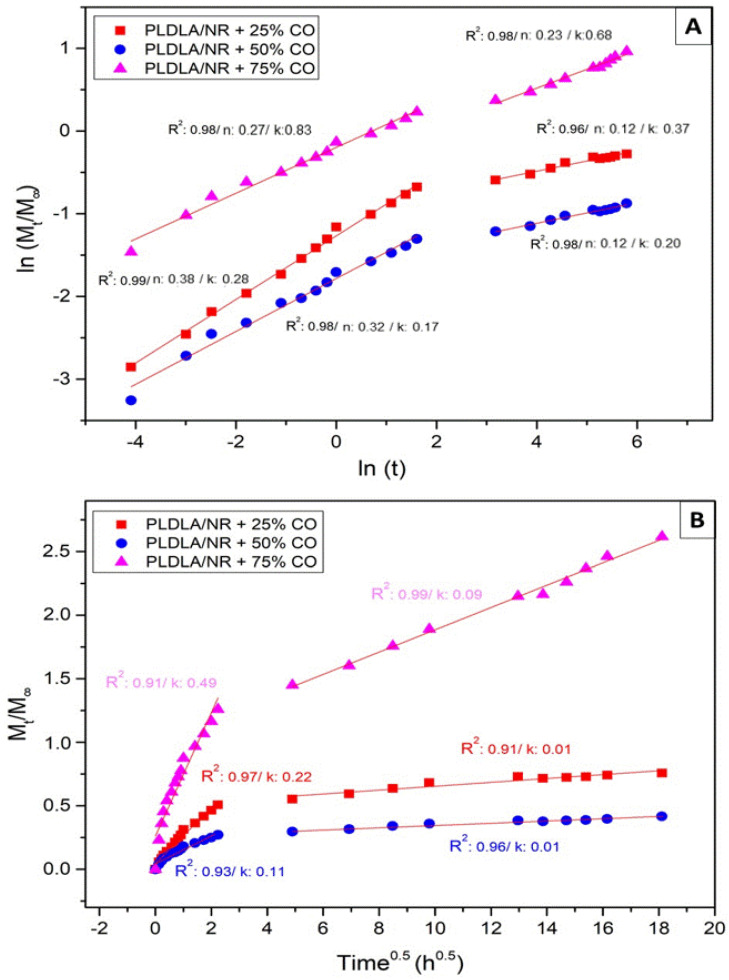
Mathematical model of CO release profile plotted using two methods (**A**) Korsmeyer–Peppas power law and (**B**) Higuchi. 2.7. swelling test.

**Figure 8 antibiotics-12-00898-f008:**
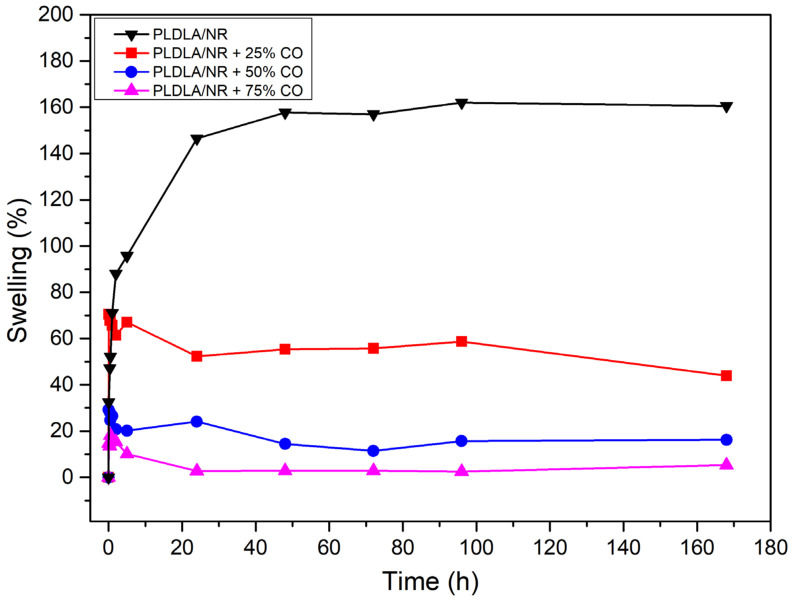
Swelling percentage of electrospun PLDLA/NR membrane and electrospun PLDLA/NR membranes with 25% CO (PLDLA/NR + 25% CO), 50% CO (PLDLA/NR + 50% CO), and 75% CO (PLDLA/NR + 75% CO).

## Data Availability

Not applicable.

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
