# Peer review of "Electrospun Membrane Based on Poly(L-co-D,L lactic acid) and Natural Rubber Containing Copaiba Oil Designed as a Dressing with Antimicrobial Properties"

_antibiotics, 2023, doi:10.3390/antibiotics12050898_

Round 1
Reviewer 1 Report
The manuscript entitled “Electrospun Membrane Based on Poly(l-co-d,l lactic acid) and Natural Rubber on Containing Copaiba Oil Designed as a Dressing with Antimicrobial Properties” by Marcelo Formigoni Pinto et al., is interesting and provides useful information about the bacteriostatic effects of the CO incorporation in combination with electrospun membranes.
The manuscript is clearly written and it is suitable for the publication, suggesting only minor revision to it.
At the end of the discussion session it could be useful to add the limitations that authors could considered in this work, such as experiments conducted on only two bacterial strains. Other bacteria including Gram positive and Gram negative related to skin infections must be considered in future work.
Also the strengths of the work could be emphasized.
Check all the manuscript: S. aureus and E. coli must be written in italic
Author Response
----------------------------------- Reviewer 1 -----------------------------------
Authors considerations: Dear Reviewer, thanks in advance for your valuable revision of our manuscript, we carefully evaluated your considerations and performed the suggested changes as described as follows:
“At the end of the discussion session it could be useful to add the limitations that authors could considered in this work, such as experiments conducted on only two bacterial strains. Other bacteria including Gram positive and Gram negative related to skin infections must be considered in future work.”
Autor responses: Based of the comment above we performed changes in text (lines 369-373) and additional explanation to justify our bacterial choice is here written as follows: As analyzed during the experimental assays, it was noted that the CO incorporated into the electrospun PLDLA/NR membranes showed excellent bacteriostatic properties only against the gram-positive bacteria S.aureus, and prevented biofilm formation. However, it showed ineffectiveness in preventing growth of gram-negative bacteria, such as E.coli, described in line 158 of the manuscript. Due to the lack of an informative bacteriostatic and bactericidal result for E. coli we depicted this second bacteria analyzed as “data not shown”. Despite, the previous and scanty data related to Copaifera officinalis L. effects, we first elected to test these 2 common bacteria, one gram positive and the other gram negative.
“Also the strengths of the work could be emphasized.”
Authors responses: Additional final remarks were presented in discussion, while assorted minor changes in text were performed.
“Check all the manuscript: S. aureus and E. coli must be written in italic.”
Authors responses: As suggested, the terms “S. aureus” and “E. coli” were checked in all manuscript and corrected to be presented in italics.
Reviewer 2 Report
The focus of the work is very interesting and shows coherence. However, there are some critical aspects that mean that the manuscript should not be considered for publication. Regarding the methodology, it is reported in sufficient detail to be replicated later by other researchers. In addition, and being the most worrisome aspect, it seems that the experiments were not carried out in triplicate, which does not give reliability. Therefore, my recommendation is to reject the manuscript.
Author Response
----------------------------------- Reviewer 2 -----------------------------------
Authors considerations: Dear Reviewer, thanks in advance for your valuable revision of our manuscript, we carefully evaluated your considerations and corrected the text with further explanations.
“…In addition, and being the most worrisome aspect, it seems that the experiments were not carried out in triplicate, which does not give reliability…”
Authors responses: Dear reviewer, during the period this project was performed, the microbiological assays were repeated independently, in different days/weeks in a total of four plates, each plate presents as indicated in figure 2, only two replicates. All independent plate presented the same presentation but we resumed in one image this results. Crytal violet assays was performed in quintuplicate as was previously indicated in line 445. However, after a critical review we observed that swelling tests and the release teste did not indicate the replicates performed which were added in this revision.
Reviewer 3 Report
The manuscript by M. F. Pinto et al reports on their recent study of antibiotic activity of membranes prepared from polylactic acid and natural rubber post-treated by Copaiba Oil. The introduction provide essential background and set up a leading hypothesis about antibiotic properties of the polymeric membranes in use as wound dressing. However, the rationale about the chosen combination of polymers is required to bring the reader to the context of this work. In general, the content of the manuscript is reasonable and main conclusion about use of the membranes as wound healing dresses are reasonable and scientifically sound. The authors are recommended to highlight the relevance of zeta-potential measurements and swelling test to the antimicrobial behavior of the membranes.
Several minor issues are to be addressed as well.
1. Title: use correct capitalization of the polymer name.
2. Abstract: "anti-biofilm formation" sounds strange. I recommend "prevention of biofilm formation"
3. Abstract and in the text: ""Crystal violet evidenced..." should be corrected, e.g. "the test with crystal violet proovs..."
line 49: too informal and vague.
Using the term "electrospinning device" applied to the product of electrospinning is incorrect.
Line 70: too informal sentence.
Line 129: "Staphylococcus aureus" is already mentioned on line 41, use "S. aureus" here and further.
In general, I support publication of this manuscript once the corrections are made.
Author Response
----------------------------------- Reviewer 3 -----------------------------------
Authors considerations: Dear Reviewer, thanks in advance for your valuable revision of our manuscript, we carefully evaluated your considerations and performed the suggested changes as described as follows:
“However, the rationale about the chosen combination of polymers is required to bring the reader to the context of this work. In general, the content of the manuscript is reasonable and main conclusion about use of the membranes as wound healing dresses are reasonable and scientifically sound. The authors are recommended to highlight the relevance of zeta-potential measurements and swelling test to the antimicrobial behavior of the membranes.”
Autor responses: Text was improved. Additional explanation related to the materials choice are described in lines 97-105. Two new references were added to support our statements. Additionally, we explain more detailed data for Zeta-potential (lines 317-323) and Swelling tests (lines 361-371).
“1. Title: use correct capitalization of the polymer name.”
Autor responses: “Electrospun Membrane Based on Poly(L-co-D,L lactic acid) and Natural Rubber on Containing Copaiba Oil Designed as a Dressing with Antimicrobial Properties”
“2. Abstract: "anti-biofilm formation" sounds strange. I recommend "prevention of biofilm formation"”.
Autor responses: In the Abstract (line 22), the term "anti-biofilm formation" has been replaced by "prevention of biofilm formation."
“3. Abstract and in the text: ""Crystal violet evidenced..." should be corrected, e.g. "the test with crystal violet proovs..."”
Autor responses: In the Abstract (line 23), the term “Crystal violet evidenced…” was replaced by “The test with crystal violet proves…”
“line 49: too informal and vague”
Autor responses: The sentence was replaced by a cohesive form (line 48-50).
“Using the term "electrospinning device" applied to the product of electrospinning is incorrect.”
Autor responses: The use of the term “electrospinning device” (line 63 and 66) was replaced by “electrospun membranes”.
“Line 70: too informal sentence.”
Autor responses: The sentence was replaced by a solemn sentence. (Line 72-75, due to previous changes). One new reference was added to support our statements.
“Line 129: "Staphylococcus aureus" is already mentioned on line 41, use "S. aureus" here and further.”
Autor responses: As suggested, "Staphylococcus aureus" was replaced by "S. aureus" (line 142).
Round 2
Reviewer 2 Report
The authors mention that the experiments were carried out in triplicate and in some cases in duplicate. However, none of the graphs show deviation bars. Also, in the discussion section, the values given are not reported as mean values with their respective standard deviation. On the other hand, another critical aspect of the work corresponds to the release of CO from the polymeric matrix, where a substantiated analysis is not made or compared with the heuristic models of drug release, which have been widely described in the field. pharmacist for several decades. In addition, it is practically an obligation to make such a comparison when working in the field of controlled drug release. I consider that these two aspects are critical and until they are delivered accordingly, the manuscript should not be considered for publication.
Author Response
Dear reviewer 2, find attached in a pdf file the responses to this second revision.
